# VectorAdam for Rotation Equivariant Geometry Optimization

**Selena Ling**
University of Toronto
selena.ling@mail.utoronto.ca

**Nicholas Sharp**
University of Toronto
nsharp@cs.toronto.edu

**Alec Jacobson**
University of Toronto, Adobe Research
jacobson@cs.toronto.edu

## Abstract

The Adam optimization algorithm has proven remarkably effective for optimization problems across machine learning and even traditional tasks in geometry processing. At the same time, the development of equivariant methods, which preserve their output under the action of rotation or some other transformation, has proven to be important for geometry problems across these domains. In this work, we observe that Adam — when treated as a function that maps initial conditions to optimized results — is *not* rotation equivariant for vector-valued parameters due to per-coordinate moment updates. This leads to significant artifacts and biases in practice. We propose to resolve this deficiency with *VectorAdam*, a simple modification which makes Adam rotation-equivariant by accounting for the vector structure of optimization variables. We demonstrate this approach on problems in machine learning and traditional geometric optimization, showing that equivariant VectorAdam resolves the artifacts and biases of traditional Adam when applied to vector-valued data, with equivalent or even improved rates of convergence.

## 1 Introduction

Over the last decades, gradient-based optimization has enabled rapid progress in machine learning and related fields. To tackle the problems caused by large variance of network weight gradients, researchers have developed adaptive variants of stochastic gradient descent such as Adam [11], which adaptively rescale the step size based on per-scalar gradient statistics to accelerate convergence.

In the field of computer graphics and geometry processing, we also see an increasing use of Adam to optimize geometric energies. For example, the progress in differentiable rendering made derivatives on rendering parameters such as geometry, lighting and texture parameters amenable to these gradient-based optimization techniques. Two recent differentiable rendering works [20, 14] default to Adam for their experiments. Recently, we also see the successful use of a customized Adam algorithm for optimizing geometric interpolation weights [31]. This work taps into this trend and takes a closer look at the use of Adam in geometric optimization problems in geometry processing and machine learning.

Another important concern in geometric learning is the design of *equivariant* functions, which have the property that rotating the input to the function causes the output to rotate in the same way. This property is natural for many geometric problems, where the choice of coordinate system is arbitrary. In this work we adopt the perspective that the optimization process is itself a function, mapping initial conditions to optimized results, and it is likewise expected that this function be equivariant. The

36th Conference on Neural Information Processing Systems (NeurIPS 2022).

widely-used Adam algorithm lacks this property, because its per-scalar statistics do not account for vector structure in the optimization variables. In particular, we show that Adam updates depend on the coordinate system chosen to represent vector-valued data, even if the minimized loss function is coordinate invariant. This leads to unexpected coordinate-dependent artifacts and biases, even when optimizing simple geometric regularization energies.

We propose *VectorAdam*, a solution that extends Adam's per-scalar operations to vectors. We test VectorAdam on a number of machine learning and geometric optimization problems including geometry and texture optimization through differentiable rendering, adversarial descent, surface parameterization, and Laplacian smoothing. We find that like regular Adam, VectorAdam outperforms traditional first-order gradient descent, yet unlike regular Adam it is provably rotation equivariant.

## 2 Related Work

Many techniques have been proposed for gradient-based optimization in machine learning and in numerical computing more broadly (see [28] for a general survey). We will primarily consider *first order* methods, which take as input only the gradient of the objective function with respect to the unknown parameters, and produce as output updated parameters after the optimization step. For large neural networks, loss gradients are often estimated stochastically on a subset of the loss function [25], however optimizers are generally agnostic to how the input gradients are computed. VectorAdam is no exception and works with stochastic samples or full gradients.

**Numerical Optimization** The most direct approach is to simply update the parameters directly via (stochastic) gradient descent, but this approach can be improved in many respects. In numerical optimization literature, techniques like momentum [18] and line search [21] offer accelerated convergence over basic gradient descent. More advanced techniques such as BFGS [8, 16] and conjugate gradient [10] make higher-order approximations of the objective landscape, while still taking only gradients as input.

**Optimization in Machine Learning** Machine learning, and in particular the training of deep neural networks, demands optimizers which have low computational overhead, scale to very high dimensional problems, avoid full loss computation, and naturally escape from shallow local minima. Here, many optimizers have been developed following the basic pattern of tracking $O(1)$ additional data associated with each parameter to smooth and rescale gradients to generate updates [6, 34, 30, 11]. All of these methods can be shown to have various strengths, but Adam [11] has emerged as a particularly popular black-box optimizer for machine learning. The crux of our work is to identify a key flaw in Adam when optimizing geometric problems in machine learning and beyond, and to present a simple resolution. We focus primarily on Adam due to its ubiquity and recent relevance in geometric optimization more broadly, but our core insights could be applied to many other gradient-based optimization schemes in much the same way.

**Variants of Adam** A number of variants have been proposed to improve Adam's performance in training neural networks. For example, NAdam [5] incorporates Nestorov momentum into Adam's formulation to improve rate of convergence, and RAdam [17] introduces a term to rectify the variance of the adaptive learning rate, among many others. Most similar to this work, Zhang et al. [37] (also published as [36]) propose ND-Adam, which preserves the direction of weight vectors during updates to improve generalization. Like our approach, Zhang et al. explicitly leverage the vector structure of optimization variables. However, their approach does so for the sake of generalization and memory savings rather than equivariance, and it produces magnitude-only updates to vectors which are unsuitable for geometric problems where points must move freely through space.

**Adam in Geometry** Recent work in 3D geometry processing has leveraged Adam to great effect, adjusting the algorithm for the problem domain. Nicolet et al. [19] proposed UniformAdam, which takes the infinity norm of the second momentum term, improving performance for differentiable rendering tasks. Wang et al. [31] customized their own Adam optimizer by resetting the momentum terms regularly in the optimization process, outperforming traditional L-BFGS on a challenging benchmark. This work aims to further this trend of Adam as a general optimization tool by modifying it to have the rotation-equivariance properties which are fundamental to geometric computing.

**Rotation Invariance and Equivariance** This work contributes to the fruitful study of invariance and equivariance in geometric machine learning [1]. When some transformation is applied to the input, an *invariant* method produces the same output, while an *equivariant* produces equivalently-transformed output. In our context, one expects that if the input to a geometric optimization problem is rotated, the output will be identical up to the same rotation—the optimizer is equivariant. Many novel invariant and equivariant layers, architectures, losses, and regularizers have been developed across geometric machine learning (see [29, 32, 7, 12, 3, 23, 35, 9, 15, 4], among many others). In this work, we argue that *optimizers* deserve the same treatment.

## 3 Background and Motivation

### 3.1 Adaptive Moment Estimation (Adam)

Given an objective function $f$ parameterized by $\theta$ and its gradient $g$, the first-order optimization algorithm Adam adaptively rescales gradients of its $t$th iteration using rolling estimations of first ($m$) and second moments ($v$) dictated by the following rules: for each scalar variable $\theta_i$,

$$
\begin{aligned}
g_i &\leftarrow \partial f/\partial \theta_i \\
m_i &\leftarrow \beta_1 m_i + (1-\beta_1)g_i \\
v_i &\leftarrow \beta_2 v_i + (1-\beta_2)g_i^2 \\
\theta_i &\leftarrow \theta_i - \alpha \left( \frac{m_i}{1-\beta_1^t} \right) \bigg/ \left( \sqrt{\frac{v_i}{1-\beta_2^t}} + \epsilon \right)
\end{aligned}
\tag{1}
$$

where $\beta_1, \beta_2, \epsilon$ are Adam's hyperparameter constants[11] and $\alpha$ is the specified learning rate.

Adam's optimization step is invariant to any uniform re-scaling of the gradients' magnitudes. The gradient is adjusted take a value of roughly $\pm 1$, bounding the effective step size by the learning rate $\alpha$. This makes it particularly suitable for problems with gradients of large variance and unknown scaling, such as neural network optimization.

### 3.2 Problems in Geometry Optimization

In recent years, we also see an increasing use of Adam in geometry optimization problems such as differentiable rendering [13, 20, 14]. In this problem domain, the target parameters are usually geometric quantities such as vertex positions of a triangle mesh, surface normal vectors, and camera positions. For all of these quantities, the vector form of their gradients encodes important geometric information that are tied to the specific coordinate system in which the problem is embedded.

Traditional Adam, originally devised for network weight optimization, does not account for the vector nature of gradients in geometry problems. The optimization process is thus not rotation equivariant—a fundamental property expected in practice—and furthermore optimization may be more unstable. Figure 1 shows an example of traditional Adam's component-wise operations creating self-intersections and axis-aligned artifacts, observed in an inverse rendering problem.



Adam    VectorAdam

Figure 1: Geometric optimization results on two sample inputs arising in a differentiable rendering task. Ordinary Adam results in axis-aligned artifacts, which are resolved by using VectorAdam instead.

When we consider gradients vector-wise, regular Adam's uniform re-scaling does not preserve the norm and the direction of the gradients. In most cases, Adam's algorithm re-scales gradient to the magnitude of $\pm 1$, restricting each optimization step within a trust region of the current parameter defined by the size of the learning rate $\alpha$ [11].

This, however, is problematic in a geometry optimization context. When all components of a gradient vector are re-scaled to be of magnitude 1, the gradient vector snaps to the diagonal direction of

the coordinate system as shown in Figure 2, changing its original direction. As shown in Figure 1, this diagonal bias in early optimization steps easily results in self-intersections near the axis and introduces geometry changes that are hard to correct in subsequent steps.

In geometry optimization problems, such as inverse rendering, self-intersections are a major source of unrecoverable failure. A non-self-intersecting surface forms the well-defined boundary of a solid shape (left in inset).Such a boundary can be displaced so that one part of the shape passes through another part, causing a self-intersection of the mesh (red in inset). Self-intersections are symptoms implying that space is no longer well separated into "inside" and "outside" (ambiguous regions shown in solid red in inset).

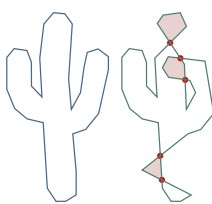

For example, consider the case where three parameters within $\theta$ form a vertex position $[x, y, z]$ in a geometry optimization problem. Denote its corresponding gradient vector $[g_x, g_y, g_z]$. The first Adam step $t = 1$ after $m, v$ are initialized with zeroes is re-written in terms of this vector-valued quantity as

$$
\begin{aligned}
[m_x, m_y, m_z] &= (1 - \beta_1)[g_x, g_y, g_z] \\
[v_x, v_y, v_z] &= (1 - \beta_2)[g_x^2, g_y^2, g_z^2]
\end{aligned}
\tag{2}
$$

and the step update for this vertex on the first iteration becomes:

$$
\begin{aligned}
\Delta[x, y, z] &= -\alpha \left[ g_x \Big/ \left( \sqrt{g_x^2} + \epsilon \right), \; g_y \Big/ \left( \sqrt{g_y^2} + \epsilon \right), \; g_z \Big/ \left( \sqrt{g_z^2} + \epsilon \right) \right] \\
&\approx -\alpha \left[ \mathrm{sign}(g_x), \mathrm{sign}(g_y), \mathrm{sign}(g_z) \right].
\end{aligned}
\tag{3}
$$

That is, for any non-vanishing gradient values, a vertex will be updated along some diagonal of the underlying coordinate system ($-\alpha[1, 1, 1]$, $-\alpha[-1, 1, -1]$, etc.). This effect creates a bias toward the corners of the coordinate system not just on the first iteration, but also throughout the optimization process (see the histograms in Figure 6 for experimental evidence).

If a gradient vector indicates how the algorithm should, for example, move a vertex in a 3D coordinate system to reach the optimal shape, the original Adam optimizer thus changes not only the magnitude but also the direction of each step. The optimization process then suffers from the loss of important geometric information encoded by the original gradient vectors' directions.

## 4  VectorAdam

We propose VectorAdam, a first-order rotation-equivariant optimizer for geometry optimizations. It tackles the aforementioned flaw of Adam for geometry optimization, by modifying Adam's per-scalar operation to operate on vectors. As shown in Figure 2, our proposed VectorAdam rescales the gradients vector-wise while preserving direction, making the optimization process rotation equivariant and stable.

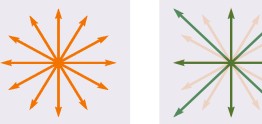
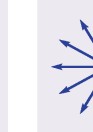

Original Gradients    Adam    VectorAdam

Figure 2: Adam's per-scalar adaptive momentum scaling results in uneven vector norm and direction inconsistency when the gradients have vector structures, while our VectorAdam scales the vector norm evenly and preserves the gradient direction.

More specifically, we note that the problems discussed above with the original Adam algorithm stem from the per-scalar calculation of the second moment $v$. VectorAdam modifies the second moment update to be the square of the *vector norm* per gradient vector instead of taking the per-scalar square. The algorithm then proceeds with the regular bias correction and updates the gradients vector-wise with corresponding products of vector first moments and scalar second moments. This has a welcome side-effect of requiring a factor of $n$ less memory for storing moments of $n$-dimensional vector parameters. Without loss of generality a description of VectorAdam is provided in Algorithm 1 assuming all parameters in $\theta$ can be arranged as an $m$-long list of $n$-dimensional vectors. If the optimization involves quantities of varying vector dimension $n$, the algorithm can be applied in tandem to each, noting that for the scalar $n = 1$ case VectorAdam reduces to ordinary Adam.

**Algorithm 1** VectorAdam: minimizing $f$ over $r$ parameters, each an $n$-dimensional vector.

---
**Require:** $\alpha$: step size
**Require:** $\beta_1, \beta_2 \in [0,1)$: Adam's exponential decay rates for the moment estimates
**Require:** $\epsilon > 0$: Adam's numerical fudge factor
**Require:** $f(\theta)$: (stochastic) objective function
**Require:** $\theta$: Initial parameter vector of shape $r \times n$
  $m \leftarrow \mathbb{0}$ (Initialize $1^{st}$ moment vector of shape $r \times n$)
  $v \leftarrow \mathbb{0}$ (Initialize $2^{nd}$ moment vector of shape $r \times 1$)
  $t \leftarrow 0$ (Initialize timestep)
  **while** $\theta_t$ not converged **do**
    $t \leftarrow t + 1$
    $g \leftarrow \nabla_\theta f(\theta)$ (Get current gradients of shape $r \times n$)
    **for** $i \in [1, r]$ **do** (Iterate through each vector quantity in rows of $\theta$)
      $\vec{m}_i \leftarrow \beta_1 \vec{m}_i + (1 - \beta_1)\vec{g}_i$
      $v_i \leftarrow \beta_2 v_i + (1 - \beta_2)\|\vec{g}_i\|_2^2$
      $\hat{\vec{m}}_i \leftarrow \vec{m}_i/(1 - \beta_1^t)$
      $\hat{v}_i \leftarrow v_i/(1 - \beta_2^t)$
      $\vec{\theta}_i \leftarrow \vec{\theta}_i - \alpha \hat{\vec{m}}_i/(\sqrt{\hat{v}_i} + \epsilon)$
    **end for**
  **end while**
  **return** $\theta$

---

Let's reconsider our 3D vertex example above, applied to the first step of optimization after initializing $\vec{m}, v$ to 0. With VectorAdam, the first, zero-initialized iteration reduces to

$$[m_x, m_y, m_z] = (1 - \beta_1)[g_x, g_y, g_z]$$
$$v = (1 - \beta_2)\,\|[g_x, g_y, g_z]\|^2 \tag{4}$$

The corresponding VectorAdam step update for this vertex becomes

$$\Delta[x, y, z] = -\alpha\,[g_x, g_y, g_z]/\|[g_x, g_y, g_z]\|\,, \tag{5}$$

that is, the unit-length direction of the vertex's gradient scaled by the learning rate $\alpha$. Thus, VectorAdam preserves the *direction* of the original gradients while roughly normalizing their *magnitude*. This achieves the vector-wise analogue of the original Adam's action on scalar-wise gradients.

## 4.1 Properties of VectorAdam

The key property which makes VectorAdam suitable for geometric optimization problem is its rotation equivariance: when the input data is rotated, the output of the optimizer will be identical up to the same rotation, at every step of optimization.

To formalize this property, we consider a function `VectorAdam` which evaluates one update step of Algorithm 1 on a single parameter vector $\theta$ with first and second moment terms denoted by $\vec{m}, v$

$$\vec{\theta}, \vec{m}, v \leftarrow \texttt{VectorAdam}_f(\theta, \vec{m}, v) \qquad \vec{\theta} \in \mathbb{R}^n, \vec{m} \in \mathbb{R}^n, v \in \mathbb{R} \tag{6}$$

for some given rotation invariant objective $f$, and internal optimizer parameters $\alpha, \beta_1, \beta_2, \epsilon$.

**Proposition 1** (VectorAdam is rotation equivariant). *For any rotation $R \in SO(n)$, let*

$$\vec{\theta}^*, \vec{m}^*, v^* \leftarrow \texttt{VectorAdam}_f(\theta, \vec{m}, v) \quad and \quad \vec{\theta}', \vec{m}', v' \leftarrow \texttt{VectorAdam}_f(R\theta, R\vec{m}, v).$$

*Then* `VectorAdam` *satisfies* $\vec{\theta}' = R\vec{\theta}^*$, $\vec{m}' = R\vec{m}^*$, *and* $v' = v^*$. *Because each step of* `VectorAdam` *is equivariant, the parameters $\vec{\theta}$ after any sequence of optimization steps are equivariant as well.*

*Proof.* Since $f$ is rotation invariant, its gradient is rotation equivariant, i.e. $\nabla f(R\theta) = R\nabla f(\theta)$. Substituting in to the update step then shows the equivariance of $\vec{m}$

$$\vec{m}' = \beta_1 R\vec{m} + (1 - \beta_1)R\vec{g} = R(\beta_1 \vec{m} + (1 - \beta_1)\vec{g}) = R\vec{m}^* \tag{7}$$

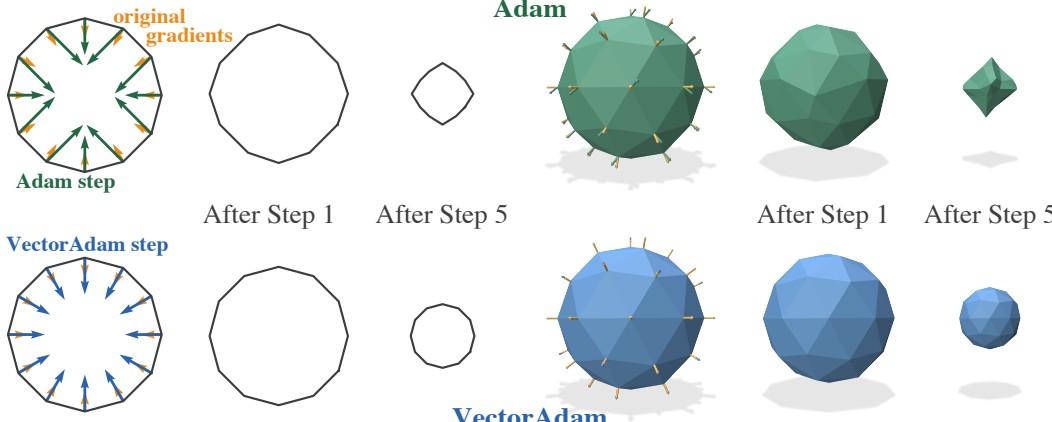

Figure 3: Laplacian regularization on 2D and 3D spherical meshes demonstrates the coordinate system bias of Adam, which changes gradients directions and magnitude breaking the intrinsic symmetry of this problem setup. VectorAdam maintains symmetry.

and likewise for the invariance of $v$

$$v' = \beta_2 v + (1 - \beta_2)||R\vec{g}||_2^2 = \beta_2 v + (1 - \beta_2)||\vec{g}||_2^2 = v^* \tag{8}$$

and the same trivially holds for their rescaled counterparts $\hat{\vec{m}}' = R\hat{\vec{m}}^*$ and $\hat{v}' = \hat{v}^*$. This then yields the equivariance of $\vec{\theta}$ as

$$\vec{\theta}' = R\vec{\theta} - \alpha\hat{\vec{m}}'/(\sqrt{\hat{v}'} + \epsilon) = R(\vec{\theta} - \alpha\hat{\vec{m}}^*/(\sqrt{\hat{v}^*} + \epsilon)) = R\vec{\theta}^*. \tag{9}$$

Induction on these relations with the base case $\vec{m} = 0$ and $v = 0$ then furthermore implies that the same equivariance holds for any sequence of successive optimization steps by `VectorAdam`.

$\square$

## 5 Experiments and Applications

We now compare the performance of Adam and VectorAdam in a range of optimization problems in machine learning and traditional geometric optimization context. These experiments demonstrate the undesirable coordinate-system bias and artifacts caused by Adam's per-coordinate update rules, and show how VectorAdam solves this problem while achieving equivalent or even improved quality and convergence rate. All experiments are performed in PyTorch on an NVIDIA RTX 2080 GPU.

Vector-valued data are especially common in geometric optimization problems, in which the gradients, also structured by vectors, carry important geometric signals for the problem at hand. We consider Laplacian regularization, adverserial descent, deformation energy optimization, and several inverse rendering tasks, all of which leverage first-order gradient-based optimization. Our VectorAdam yields stable and rotation equivariant optimization, and often arrives at a better result by preserving the geometric signals encoded by the gradient vectors.

**Laplacian Regularization**

### 5.1 Geometric Optimization

Laplacian regularization on a mesh is widely-used in geometric problems, like 3D reconstruction and generative modeling tasks, especially to discourage mesh self-intersections. Applicable to any mesh or graph problem (e.g., point clouds + $k$ nearest neighbors), this loss term adds a smoothness prior on 3D positions by requiring them to be as similar as possible to their neighrbors'. For surfaces this loss is often a discretized version of Dirichlet energy ($\int ||\nabla x||^2$), expressed in the form of $vLv^T$ in which $v$ is the mesh vertices and $L$ represents the Laplacian matrix. It is intrinsic and does not depend on the rotation of the shape.

In this experiment, we use both Adam and VectorAdam to directly minimize the Laplacian regularization loss on a mesh. We observe that Adam's per-coordinate moment updates introduces undesirable axis-aligned distortion to the Laplacian regularization. In Figure 3, Adam's optimization steps shrink the original sphere in 3D, or circle in 2D, to a diamond shape after a few optimization steps, which aligns with the diagonal bias we discussed before. In addition, in Figure 4, we show that Adam's gradient update is not rotation equivariant, resulting in a different updated mesh given a rotated input mesh. In contrast, VectorAdam follows the geometric intuition and shrinks the sphere, or circle, smoothly across the surface in Figure 3 and updates the rotated input mesh identically in Figure 4.

**Deformation** We also experiment with two common deformation loss optimizations in 2D. These energies can be seen as priors defining the smooth deformation. In particular, we minimize the as-rigid-as-possible energy and the symmetric Dirichlet energy on a deformed 2D mesh given a reference mesh.

The as-rigid-as-possible energy measures how much a mesh has become distorted relative to an initial "rest" configuration. Given an edge connecting $p_i$ and $p_j$ in the original mesh, $p'_i$ and $p'_j$ in the deformed mesh, the as-rigid-as-possible energy at this edge is defined as $\min_{R_{ij}} ||(p_i - p_j) - R_{ij}(p'_i - p'_j)||$ with the best fit rotation $R_{ij} \in SO(3)$. This energy is interesting because it measures the local stretching and squashing of the triangles while factoring out *local* rigid motion [27]. It is a commonly-used deformation energy with its detail-preserving property.

We optimize the ARAP energy for a 2D triangle mesh and visualize the per-vertex gradient vectors for the first optimization step in the top left

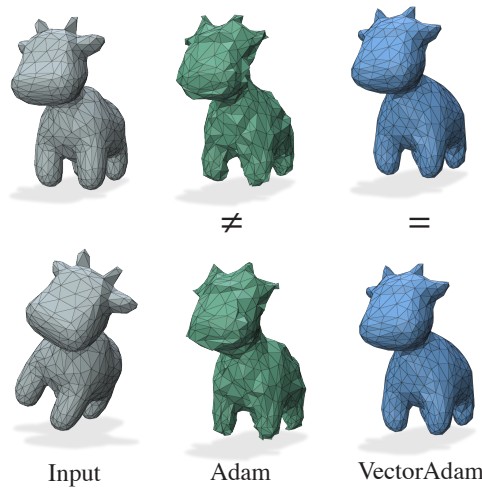

Input          Adam          VectorAdam

Figure 4: We optimize for Laplacian smoothing loss on a canonical mesh and a rotated mesh with both Adam and our VectorAdam. The output of ordinary Adam is significantly altered by the rotation, while our VectorAdam yields an identical update up to rotation, as expected.

panel of Figure 6. Gradient Descent has no coordinate-system bias, but wide range in step scales. If the gradient vector for a particular vertex is $[x\,y]$, then the first VectorAdam step will normalize this to $-\alpha\widehat{[x\,y]}$. Accordingly, these uniform-length steps show no coordinate-system bias. Meanwhile, the first step of Adam will take a step of $-\alpha[\text{sign}(x)\,\text{sign}(y)]$, thus exhibiting extreme bias to the $45°$ directions.

Symmetric Dirichlet energy is an alternative way of measuring triangle distortions between a deformed mesh and the reference mesh. It is based on a computationally efficient form of isometric distortion metric that uses the eigenvalues of the affine transformation between two triangles. Optimizing the Symmetric Dirichlet energy prevents local folds of triangles. Similar to the Laplacian regularization, the energy is intrinsic to the mesh, and therefore rotation invariant. We optimize for symmetric Dirichlet energy 16 times with the input mesh rotated by a different angle and plot the loss curves in Figure 5. As shown in the Figure 5, VectorAdam follows the same optimization path regardless of the starting orientation, whereas Adam produces different optimization paths. VectorAdam achieves equivalent convergence rate and final quality as traditional Adam.

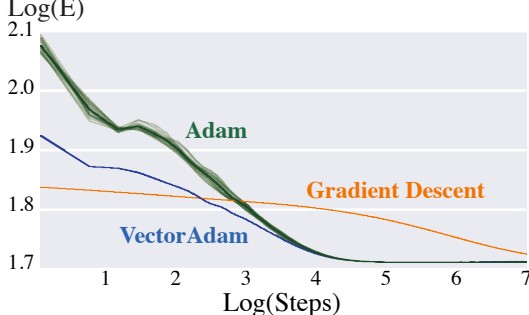

Figure 5: Log loss curves with symmetric Dirichlet energy [26] optimization with rotated inputs. We show that our VectorAdam follow the same descent path as desired while Adam's descent path varies across rotated inputs.

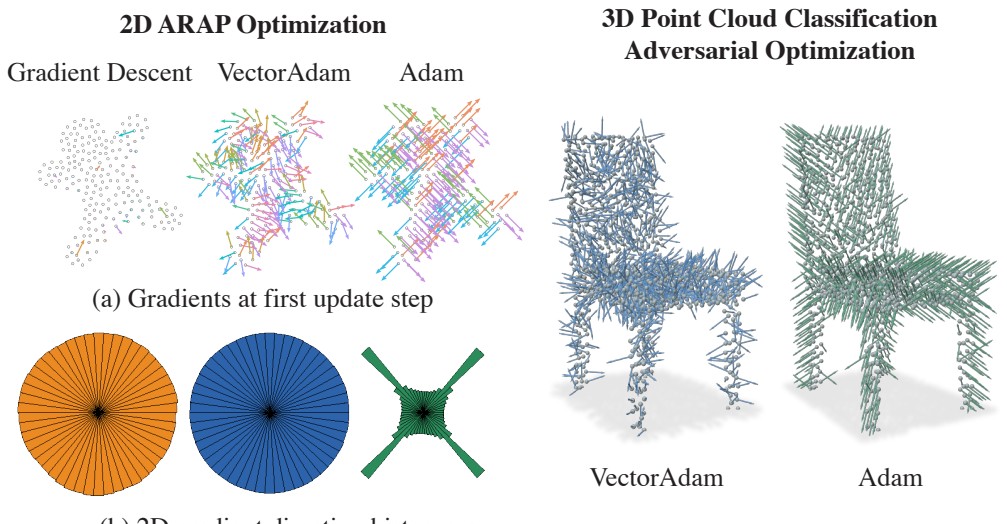

**2D ARAP Optimization**

Gradient Descent    VectorAdam    Adam

(a) Gradients at first update step

(b) 2D gradient direction histogram

**3D Point Cloud Classification Adversarial Optimization**

VectorAdam      Adam

Figure 6: On the left, we optimize the ARAP energy for a triangle mesh with 200 vertices for 100 iterations with 1000 randomly rotated initial configurations. (a) shows the gradients at first optimization step from Adam and VectorAdam. (b) shows the histograms of the angular distribution of these 20M gradient directions. Gradient Descent and VectorAdam exhibit their rotation equivariance by presenting a uniform distribution, while Adam presents a biased distribution with spikes at $45°$ angles. On the right in (c), we observe similar coordinate-system bias on a input point cloud, when adverserially optimize inputs to a 3D classifier.

**Adversarial Descent**    Similar challenges occur in the context of geometric machine learning. One such example is adversarially optimizing input data to minimize or maximize a classifier's prediction, a process which is vulnerable to Adam's coordinate bias. Here, we trained a point cloud classifier from the rotation-equivariant Vector Neuron architecture [4] on the ModelNet40 dataset [2], and adversarially optimize an input point cloud to minimize the classifcation loss. In Figure 6, *right*, Adam's gradient updates exhibit the diagonal bias while VectorAdam preserves the directions.

**Inverse Rendering**    Research progress in differentiable rendering in recent years has made image-based inverse geometry optimization a popular problem. Within this problem domain, we experiment with shading-based

inverse geometry and texture reconstruction. Both of these problems optimize vector-valued parameters, i.e. vertex positions and RGB color vectors.

For geometry reconstruction, we run the same experiments as in [19], which optimize a sphere to match against rendering of a Nerfertiti bust. More specifically, we run the optimization 8 times with different learning rates in the range of 9e-6 and 1e-5, and show the Hausdorff distances between the optimized mesh and the target mesh

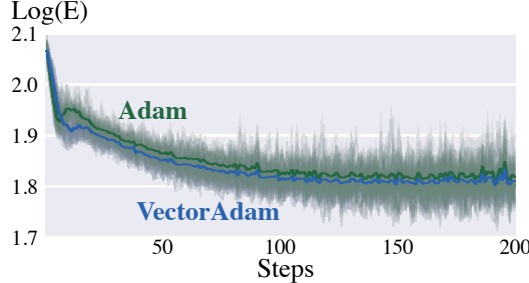

Figure 7: Log loss curves for environment map reconstruction with Adam and VectorAdam, over 200 runs. The idea of VectorAdam also benefits optimization of non-geometric quantities that have vector structures such as RGB colors.

in Figure 8 during optimization. Quantitatively, VectorAdam's optimized mesh consistently achieves better, i.e. lower, Hausdorff distance with the reference mesh. Qualitatively, we observe more self-intersecting faces in Adam's optimized mesh as highlighted in red.

We note that an Adam variant called UniformAdam was proposed in [19], which also preserves gradient direction by taking the infinity norm of the second moment estimation. However, UniformAdam is

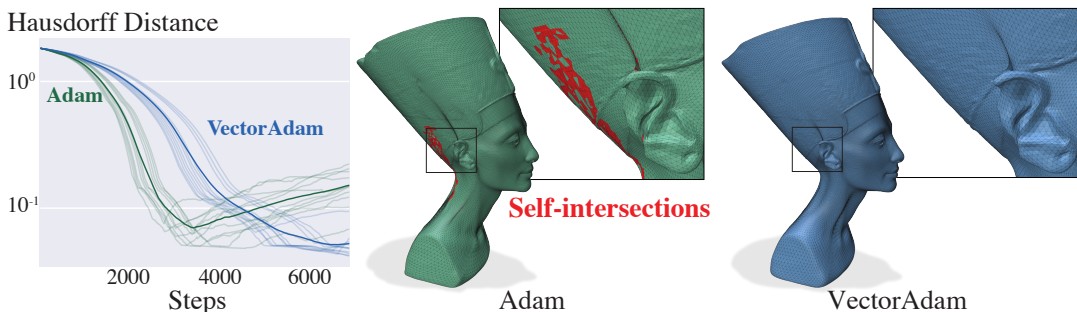

Hausdorff Distance

Adam    VectorAdam    Self-intersections

Adam    VectorAdam

Figure 8: For a shading-based inverse reconstruction of the Nefertiti bust, Hausdorff distance between optimized mesh and target mesh across multiple runs with learning rates in the range of [9e-6, 1e-5] shows that VectorAdam consistently outperforms Adam. We observe that qualitatively Adam more often results in tangled surface self-intersections which are difficult to recover from (red), whereas this VectorAdam result does not suffer from this problem.

not rotation equivariant as the per-coordinate infinity norm changes under rotation. Our VectorAdam can easily be modified in a similar way while preserving the property of rotation equivariance by taking the infinity norm over gradient vector norms in the second moment estimation.

In addition, we also extend our experiment to non-geometric vector-valued data with the problem of texture reconstruction. We run the same texture reconstruction script, as provided by [13], 200 times with different initializations. In Figure 7, we show VectorAdam consistently arrives at better texture result while achieving better convergence rate. This shows that our insight on formulating optimizer by vectors can also benefit other non-geometric problems with vector-valued data. In future work, we are interested in investigating the meaning of rotation in-/equi-variance in color space.

## 6 Future Work

The above experiments focus mainly on coordinate-valued geometric data, where the properties of VectorAdam are most easily evident and well-founded in theory (Section 4.1). However, preliminary experiments also show promise for VectorAdam in the more general context of network weight optimization in geometric machine learning.

A classic task in geometric machine learning is point cloud segmentation. We perform a preliminary experiment using the widely-adopted PointNet architecture [24], in which we train the network using both VectorAdam and conventional Adam on a subset of ShapeNet[33] dataset. More precisely, we apply VectorAdam's equivariant update to the rows of PointNet kernel gradients. For example, consider the kernel weights in the first convolutional layer in the shape of $n \times 3$, which transforms the 3-dimensional point cloud coordinate data into $n$-dimensional latent vectors; our VectorAdam preserves the direction of each row vector in the

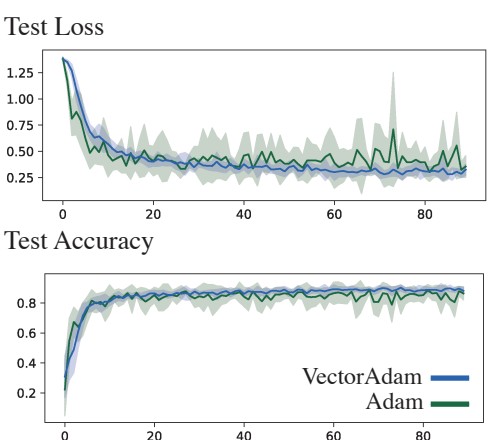

Figure 9: Average test loss and accuracy over 10 runs for point cloud segmentation using PointNet, trained with VectorAdam and Adam. The shaded region indicate the area between plus and minus one standard deviation of test losses and accuracies over 10 runs.

kernel. In Figure 9, we plot the test accuracy and loss throughout the training process, and observed a significant increase in stability using VectorAdam.

These network weights are not coordinate-valued, but can be potentially interpreted geometrically as they map the point coordinates into latent spaces to perform certain geometry-aware tasks such as shape classification and segmentation. Therefore, this preliminary experiment points to the potential benefits of geometry-aware optimization in the broader machine learning context. However, this observed stability is not fully understood in theory, and we hope to investigate it further in the future.

# 7 Conclusion

Our proposed VectorAdam alleviates the identified coordinate-system dependency flaw of Adam. We proved rotation equivariance of VectorAdam theoretically and demonstrated its implications for popular geometric optimization tasks in geometry processing and machine learning more broadly.

We suspect that networks which take positional coordinate values as input but whose degrees of freedom are layers of weight matrices (e.g., [22]) may implicitly suffer from the same lack of rotation equivariance. Unfortunately, our VectorAdam is only well understood for problems where the degrees of freedom are explicitly vector-valued quantities such as positional coordinates. Neural networks which take such coordinates as input but whose degrees of freedom are layer weight matrices may also suffer from coordinate-dependence bias under the traditional Adam optimizer. Although our preliminary PointNet experiment shows an increase in test-time stability with VectorAdam, it is not yet clear how the rotational equivariant property of VectorAdam will directly apply to these network weight optimization.

Additionally, the theoretical underpinnings for Adam have primarily been investigated in the stochastic optimization case; some of the direct geometric tasks considered here would benefit from more analysis to understand why Adam and VectorAdam prove so effective. In this work, we have focused on rotations and low-dimensional problems in geometry processing. In future work, we are further interested in investigating invariances and equivariances to other transformation groups and high-dimensional vector or tensor spaces.

We hope that our attention to rotation equivariance for *optimizers* encourages further development of coordinate-free geometric optimization within the evolving machinery of continuous optimization and deep learning.

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
