# OpenReview forum: "VectorAdam for Rotation Equivariant Geometry Optimization"
_NeurIPS.cc/2022/Conference — NeurIPS 2022 Accept_

### Official Review · Reviewer_KcD5 · 2022-07-09

**Rating:** 4
**Confidence:** 3
**Soundness:** 2 fair
**Presentation:** 3 good
**Contribution:** 2 fair

**Summary:**

The authors propose a variant of the gradient-based optimization algorithm Adam, which they denote VectorAdam, which is equivariant with respect to rotation of vector valued data. This is realised by changing the Adam gradient and second moment computations, which assume each parameter is a scalar quantity, with the appropriate quantities for vector valued parameters.

The authors show in some geometric optimization experiments that VectorAdam is indeed rotationally equivariant, while Adam is not. Further experiments are performed, with the goal of highlighting advantages of VectorAdam: for example, in the task of inverse rendering, they show that Adam gives rise to self-intersecting surfaces, while VectorAdam does not.


**Questions:**

A major improvement to VectorAdam that I can see make it a much more impactful paper would be to extend the algorithm for equivariant optimization of any representation of SO(n). Such representations are part of layers of recent equivariant neural networks (eg, e3nn). Thus, this would also apply to networks that take geometric coordinates as inputs as well.


**Limitations:**

The authors describe quite well the limitations of their approach.

Overall, I think the work is worth of publication in a some more technical venue, but it does not achieve the level of novel and impact to the machine learning community to justify the publication in a venue like NeurIPS. I would encourage the authors to extend the scope of applicability of VectorAdam to, perhaps, TensorAdam, which could cover the full extend of equivariant networks.


**Strengths And Weaknesses:**

The paper is clear and well written. The assumptions are clearly stated and the algorithm is clearly described. The paper addresses an interesting topic that deserves attention. Especially since the recent popularity of equivariant neural networks, which often are difficult to train, having geometry aware optimisers is likely important for the exploiting the full potential of such models.

The main weaknesses of the paper consists in the fact that it introduces just a simple variation with respect to the original Adam algorithm, where the gradients for a vector are computed instead of scalar parameters. The experiments substantially provide evidence of the claim of rotation equivariance of VectorAdam. Other than that, I’m not sure that such machine learning techniques are relevant enough in the machine learning community to justify the relevance of substantially just an incremental modification of the (very successful) Adam optimizer. In fact, as Figure 5 shows, it can be seen that in some of these optimization problems Adam finds the same minimum as VectorAdam.

---

> ### Author Response · Authors · 2022-08-02
> **Response to Reviewer KcD5**
>
> Thank you for the comments.
> > “simple variation”, “I’m not sure that such machine learning techniques are relevant enough in the machine learning community to justify the relevance of substantially just an incremental modification of the (very successful) Adam optimizer.”
>
> We consider the simplicity of this method to be a valuable strength, rather than inventing an entirely new optimizer, we are able to offer the desired equivariance properties while building on an optimizer which has been widely studied and adopted within the community. This will facilitate uptake and increase impact of our method. Also, please see our General Response 1 for our work’s relevance to the NeurIPS community.
>
> > “major improvement to VectorAdam that I can see make it a much more impactful paper would be to extend the algorithm for equivariant optimization of any representation of SO(n). Such representations are part of layers of recent equivariant neural networks (eg, e3nn).”
>
> This is a great idea! We see deeper generalization of the basic technique introduced here as interesting potential future work, though our experiments (and the interest of other reviewers) already show that our treatment of the rotation-equivariant vector case has value for the community. Also, please see our General Response 2 for a general discussion on extending VectorAdam to network weights.

---

> > ### Comment · Reviewer_KcD5 · 2022-08-09
> > **answer**
> >
> > I thank the authors for their reply.
> >
> > I understand that simplicity is often a strength, and unnecessary complexity is not a desired feature. However, I am still of the opinion that the novelty of the proposed methodology is too limited, as well as its filed of applicability. Not the topic in itself, so previous NeurIPS publications in the field are not an argument for the current paper in my opinion. As I mentioned before, I think a true VectorAdam should be able to deal with both scalar and vector fields as well as higher representation of SO(n), and I do not think this should be the topic of future works but it would fit very well in this one.
> >
> > You also mention you perform an experiment on PointNet. Where is it to be found? As a reviewer I can only consider an experiment if it is included in the revised draft.
> >
> > I am keeping my vote for now, but I'm looking forward to the discussion with the other reviewers.

---

### Official Review · Reviewer_oBCs · 2022-07-11

**Rating:** 8
**Confidence:** 3
**Soundness:** 3 good
**Presentation:** 4 excellent
**Contribution:** 4 excellent

**Summary:**

The paper suggests a vector-modified version of the popular Adam optimization algorithm. The motivation stems from problems where the optimization variables are coordinates positions and the underlying optimized energy is rotation invariant, a typical setting in geometry processing. The main idea is to consider gradient update steps consisting of vectors (instead of scalars) and modify momentum normalization such that the results step is rotation equivariant.
The method is evaluated on various geometry processing tasks, showcasing its merits.


**Questions:**

Please address the weakness stated above.

Some interesting questions I would have liked the paper to address are:
Are there any limitations to constraining the optimization solution space? Are there settings where it fails? Does it require a more careful initialization design?

In addition, for not geometry processing experts, it would be beneficial to elaborate on the self-intersection issues and the red lines in figures 1 and 8.


**Limitations:**

The paper does not include such a discussion.

**Strengths And Weaknesses:**

Strengths
-------------
The paper is very well written and easy to follow.
The background and motivation section does a great job of overviewing some of the issues of Adam in geometry optimization.
The proposed method is simple yet seems to be effective. It feels like a good starting point to encourage further research.
The method is evaluated on an adequate range of problems, showing improved qualitative results.


Weaknesses
----------------

The paper could better explain the role of equivariance in optimization settings versus the known settings used in modeling functions (networks). I was a bit confused that in contrast to equivariant networks where it is also valuable to constrain the prediction of transformed inputs (the prediction of a rotated point cloud is predictable), here, the inputs to the problem are not expected to be transformed. It is just that by using the equivariant constraint, the class of possible solutions is reduced (which is a strong and useful inductive bias to rotation invariant energies).

Moreover, I think it would be better to define equivariant already at the beginning of the paper, including a statement that the paper assumes rotation invariant energies.

The evaluation should include quantitative results. Does the VectorAdam local minimum is lower/higher than Adam solutions (although qualitatively better).

---

> ### Author Response · Authors · 2022-08-02
> **Response to Reviewer oBCs**
>
> Thank you for the comments.
> > “better explain the role of equivariance in optimization settings versus the known settings used in modeling functions (networks).”
>
> Please see General Response 2 for such discussion. We also find it useful to consider the problem from a functional perspective. Typically, equivariance is studied in the context of a neural network as a function which maps some input to some output in the same domain; that function should be equivariant. In this work, we note that the optimization process is also a function which maps some input data to some output, and it is thus valuable to seek and study fully equivariant optimization processes.
>
> > “I think it would be better to define equivariant already at the beginning of the paper, including a statement that the paper assumes rotation invariant energies”
>
> Good point! We have updated the paper in the introduction to include this (see green text in paper). We also refer to 4.1 for a formal mathematical proof of VectorAdam’s rotation equivariant property.
>
> > “The evaluation should include quantitative results. Does the VectorAdam local minimum is lower/higher than Adam solutions (although qualitatively better).”
>
> Please see quantitative results in Figure 7 and 8. We do observe that on some problems the resulting minima found by VectorAdam are indeed superior, and in our experiments they were never significantly worse.
>
> > “Are there any limitations to constraining the optimization solution space?”
>
> To be clear, we are not constraining the optimization solution space—the degrees of freedom for the problem are unchanged, and we do not constrain the gradient update direction. However, our changes affect Adam’s update scaling, which means that optimization trajectories are indeed affected. In particular, we guarantee that if the objective is _invariant_, then the optimization trajectory is _equivariant_ with respect to the input parameters. When optimizing objectives with many local minima, a different trajectory may of course lead to different solutions; often better ones!
>
> > “Are there settings where it fails?”
>
> There are settings where rotation equivariance is not desirable. For example, in Latin character recognition, the letter ‘p’ looks like a rotated letter ‘d’. Likewise, one might imagine settings where the desired minima of an optimization are not necessarily rotation-equivariant, such as biasing results along the upward (Z) dimension due to the natural gravitational orientation of 3D objects. (Although in this example, we would suggest applying VectorAdam treating only the XY dimensions as vectors instead.)
>
> > “Does it require a more careful initialization design?”
>
> We did not use any special initialization strategies in our experiments, and there is no reason why they should be necessary. However, this is a very interesting question to consider more generally: how can one design vector-aware initialization strategies for geometric learning problems, and what benefits do they offer? Such work would nicely complement our own study of optimizers.
>
> > “In addition, for not geometry processing experts, it would be beneficial to elaborate on the self-intersection issues and the red lines in figures 1 and 8.”
>
> Good point! We have updated Section 3.2 to include more discussions and an illustrative diagram (see green text in paper). For certain geometric optimization scenarios, the self-intersections at the first step would introduce geometry changes that are hard to correct in subsequent optimization steps, resulting in irreversible artifacts.

---

> > ### Comment · Reviewer_oBCs · 2022-08-09
> > **answer**
> >
> > I thank the authors for their detailed answers.
> >
> > The functional perspective on optimization is nice. I encourage the authors to further formalize it and incorporate it in the paper as well.
> >
> > I reviewed the revised changes. To my opinion, the changes improve the paper. Please note the added typo in line 115.
> >
> > To summarize, I will keep my original score, thanks.

---

### Official Review · Reviewer_2Wxe · 2022-07-11

**Rating:** 7
**Confidence:** 4
**Soundness:** 4 excellent
**Presentation:** 4 excellent
**Contribution:** 3 good

**Summary:**

The Adam optimization algorithm has proven effective on tasks in geometric optimization we often have vector-valued data). However, the updates we obtain by naively applying Adam does not preserve the original gradient direction (due to magnitude rescaling), and the Adam updates are coordinate dependent (hence not rotation equivariant). This paper proposes *VectorAdam* which preserves the gradient direction and is provably rotation equivariant. The authors also demonstrate empirically on a variety of geometric optimization tasks that, the proposed modification avoids artifacts and biases created by the ordinary Adam, with equivariant or even improved rate of convergence.

**Questions:**

1. Why Adam optimization is a good algorithm for geometric optimization? Is it significantly better than plain gradient descent (which of course preserves the original gradient)? If true, why that's the case? Do the authors have some intuition about this ?
2. Neural implicit representation is an emerging area of research with lots of applications, especially inverse rendering. It has a neural network that takes in positional coordinates. Do you think the proposed VectorAdam can be useful?

[1] Implicit Neural Representations with Periodic Activation Functions (https://arxiv.org/abs/2006.09661)
[2] Representing Scenes as Neural Radiance Fields for View Synthesis (https://arxiv.org/abs/2003.08934)

**Limitations:**

I have no concern in terms of potential negative societal impact.

**Strengths And Weaknesses:**

**Strengths**
1. The authors provide a good observation that, while Adam is effective in geometric optimization, it will create biases and artifacts, and the optimization results are coordinate dependent (which is not ideal in the setting of geometric optimization). They propose a simple yet effective modification that solves the issues abovementioned.
2. All claims in this paper are backed by theoretical proof and sufficient empirical study.
3. The paper is very well-written, with illustrations and simple examples for clarity. It is a pleasure to read this submission.
4. Geometric optimization is a useful application that may appear in computational chemistry, physics, computer vision etc. This work provides a modification to the Adam optimization algorithm which has been previously applied to this problem successfully. I believe this work is very useful in related areas of research, and may inspire future research on the underexplored equivariant optimization algorithms.

**Weaknesses**
1. This work may become a stronger submission if they can provide more analysis regarding why the artifacts and biases will appear in addition to their visual examples. A mathematical quantification and analysis would be good because it may shed some light on future development beyond vector-valued quantities.
2. The paper might benefit from having more content, such as an exploration of equivariance in colour space (as they mentioned), or explain more about geometric optimization to people who are not familiar with this special type of optimization problem. Currently the paper is slightly more than 8 pages and they have enough space.

---

> ### Author Response · Authors · 2022-08-02
> **Response to Reviewer 2Wxe**
>
> Thank you for the comments.
>
> > “more analysis regarding why the artifacts and biases will appear in addition to their visual examples”
>
> We’ve added discussions in Sec 3.2 to clarify (see the green text). In brief, for certain geometric optimization scenarios, the self-intersections at the first step would immediately introduce geometry changes that are hard to correct in subsequent optimization steps, therefore resulting in irreversible artifacts.
>
>
> > “A mathematical quantification and analysis would be good.”
>
> We provide a mathematical proof for VectorAdam’s rotation equivariance property in Section 4.1. We omitted a formal proof for Adam’s lack of rotation equivariance because we felt it was already directly clear. We are happy to include a more formal proof if it helps. In addition, we point to Figure 6(b) for a visual quantification of the bias introduced by Adam in an as-rigid-as-possible deformation problem.
>
> > “exploration of equivariance in color space”
>
> Our experiment in Figure 7 with color optimization explores the benefits of VectorAdam on a non-geometric form of vector-structured data. Although VectorAdam converges to a better optimum, it’s unclear just what the implications are of equivariance in color space; this is a nuanced topic for which we mostly defer to the color science community; for example, rotations about the hue axis may be meaningful, whereas rotations that change luminance may not be.
>
> > “or explain more about geometric optimization to people who are not familiar with this special type of optimization problem”
>
> We have updated the paper, in Section 5.1 under Laplacian Regularization and Deformation (see green text in paper), to include more detailed explanations of the geometry optimization problems.
>
> > “Why Adam optimization is a good algorithm for geometric optimization? Is it significantly better than plain gradient descent (which of course preserves the original gradient)? If true, why that's the case? Do the authors have some intuition about this ?”
>
> Good question! It is indeed better than plain gradient descent, and we refer to Figure 5 for one of the experiments that compared against plain gradient descent. [1] is also an example of recent work in which a modified Adam outperformed plain gradient descent. Meanwhile, Adam outperforms stochastic gradient descent for inverse rendering problems and is the conventional default (e.g., [2], [3], [4], [5], [6]).
> More generally, gradient descent is coordinate system independent but lacks any momentum. From this perspective, VectorAdam retains coordinate independence but introduces a momentum term, and observationally leading to performance similar to Adam. However, there is not yet a complete theory behind why momentum-based optimizers such as Adam work so well for geometric optimization problems; we hope our results can inspire future work looking into this question.
>
> > “Neural implicit representation is an emerging area of research with lots of applications, especially inverse rendering. It has a neural network that takes in positional coordinates. Do you think the proposed VectorAdam can be useful?”
>
> Yes! This is definitely an exciting area for future investigation. Please see the discussion in General Response 2, including particular works such as [7] and [8] that could potentially benefit from an optimizer like VectorAdam.
>
> **Citations**
>
> [1] Wang, Yu, and Justin Solomon. "Fast quasi-harmonic weights for geometric data interpolation." ACM Transactions on Graphics (TOG) 40.4 (2021): 1-15.
>
> [2] Nicolet, Baptiste, Alec Jacobson, and Wenzel Jakob. "Large steps in inverse rendering of geometry." ACM Transactions on Graphics (TOG) 40.6 (2021): 1-13.
>
> [3] Chen, Wenzheng, et al. "Learning to predict 3d objects with an interpolation-based differentiable renderer." Advances in Neural Information Processing Systems 32 (2019).
>
> [4] Nimier-David, Merlin, et al. "Mitsuba 2: A retargetable forward and inverse renderer." ACM Transactions on Graphics (TOG) 38.6 (2019): 1-17.
>
> [5] Laine, Samuli, et al. "Modular primitives for high-performance differentiable rendering." ACM Transactions on Graphics (TOG) 39.6 (2020): 1-14.
>
> [6] Liu, Shichen, et al. "Soft rasterizer: A differentiable renderer for image-based 3d reasoning." Proceedings of the IEEE/CVF International Conference on Computer Vision. 2019.
>
> [7] Atzmon, Matan, et al. "Controlling neural level sets." Advances in Neural Information Processing Systems 32 (2019).
>
> [8] Williams, Francis, et al. "Neural splines: Fitting 3d surfaces with infinitely-wide neural networks." Proceedings of the IEEE/CVF Conference on Computer Vision and Pattern Recognition. 2021.

---

> > ### Comment · Reviewer_2Wxe · 2022-08-05
> > **Convincing rebuttal response**
> >
> > The authors did a great job and answered all my questions convincingly. They have also updated their manuscript to implement the suggestions. I will keep my original score.

---

### Official Review · Reviewer_5HeJ · 2022-07-13

**Rating:** 5
**Confidence:** 4
**Soundness:** 3 good
**Presentation:** 4 excellent
**Contribution:** 1 poor

**Summary:**

In this work, the authors present a vector-based version of the Adam optimizer called VectorAdam. In contrast to Adam, VectorAdam operates on the vector form of the gradients instead of performing the scaling on the independent values within the gradient vector. This conceptual change primarily results in the update to the second-order moment no longer using the square of the individual values but using the norm of the vector instead. The authors show that with this change, VectorAdam will have rotation equivariant weight updates.

The authors evaluate their proposed optimizer experimentally for experiments; Laplacian Regularisation, Deformation, Adversarial Decent, and Inverse rendering. In all three approaches, VectorAdam shows a benefit over regular Adam.

**Questions:**

As stated above, the main question arising from the paper relates to the relevance of the proposed optimizer to the machine learning community. As it stands, it is unclear how weight-equivariance relates to output-invariance. This should be clarified in the rebuttal.

**Limitations:**

The paper is accompanied by a clear code base. Aside from the issues discussed above, there are no further limitations that should have been highlighted in the paper.

**Strengths And Weaknesses:**

The presented paper is well-written, contains an extensive related work section and presents a number of interesting experimental evaluations.

However, based on the current version of the manuscript it is unclear to what extent the proposed VectorAdam optimizer is relevant to the NeurIPS community. Currently, only the adversarial descent experiment is related to machine learning. However, also here the setup is different from the main use case of Adam in machine learning—updating the weights of the model—and instead focuses on determining an adversarial update to the point could points.

More generally, the main weakness of the paper is that it is unclear how output-space invariance (as we are mostly interested in machine learning) translates to equivariance in the weight-space. Without this clarification, the paper might be more suitable for a  differential geometry-specific venue.

In addition to that, I would like to highlight some minor comments:
- In both the definition of Adam and VectorAdam, the initial m_i should be m_{i-1} before the update.
- In the algorithm, both the shape of the parameter vector and the first order moment uses $m$ as a variable. This is slightly confusing. To stay in line with previous work, I would suggest changing the notation for the shape of the parameter vector.

---

> ### Author Response · Authors · 2022-08-02
> **Response to Reviewer 5HeJ**
>
> Thank you for the comments.
>
> > “It is unclear to what extent the proposed VectorAdam optimizer is relevant to the NeurIPS …
> >“It is unclear how output-space invariance (as we are mostly interested in machine learning) translates to equivariance in the weight-space.”
>
> First, we wish to re-emphasize the discussion from General Response 1 above: there are many topics beyond the weights of neural networks which are of great interest and relevance to the NeurIPS community!
>
> That being said, perhaps it is useful to consider the problem from a functional perspective. Typically, equivariance is studied in the context of a neural network as a function which maps some input to some output in the same domain; that function should be equivariant. In this work, we note that the optimization process is also a function which maps some input data to some output, and it is thus valuable to seek and study fully equivariant optimizers. Under this perspective, input/output space equivariance of the optimization process is very relevant.
>
> See also the discussion of a preliminary experiment in weight optimization in General Response 2.
>
> > “In both the definition of Adam and VectorAdam, the initial $m_i$ should be $m_{i-1}$ before the update.”
>
> Here, the index $i$ identifies the $i$th optimization variable. $m_i$ refers to the rolling moment update for the target scalar or vector variable for the $t$th gradient descent step.
>
> > “In the algorithm, both the shape of the parameter vector and the first order moment uses m as a variable. “
>
> Thank you for pointing out the confusing use of m. We have updated the paper in the algorithm to use a different notation.

---

> > ### Comment · Reviewer_5HeJ · 2022-08-04
> > **Response**
> >
> > Thank you for your reply. First and foremost, I must state that I agree with the reviewers that there are other concepts relevant to the NeurIPS community beyond that of updating neural network weights. However, the authors start their abstract by stating "the rise of geometric problems in machine learning", followed by separating the concepts of geometric optimisation and machine learning, and ultimately, conclude the abstract by stating that they apply their proposed approach to problems in machine learning. These statements do warrant a more in-depth discussion of the relevance of VectorAdam for machine learning beyond the single adversarial perturbation experiment.
> >
> > If the authors could provide more information about the additional experiment mentioned in general response 2 this would be very helpful.
> >
> > Thank you for your further clarifying comments.

---

> > > ### Author Response · Authors · 2022-08-06
> > > **Response**
> > >
> > > ### On Abstract
> > > ____
> > > Thank you for pointing out the ambiguous wording in our abstract. As mentioned in our General Response 1, we believe our method applies to coordinate-valued geometric data within traditional geometry processing tasks, as well as the machine learning pipeline, which is what we intend by “geometric problems in machine learning”. In addition to the adversarial optimization task, the differentiable rendering and the laplacian regularization task are both common modules in the geometric machine learning pipeline. Increasingly, these domains are converging, as learning techniques become more deeply integrated into 3D tasks. Therefore, we are happy to update our abstract as below:
> > >
> > > **The Adam optimization algorithm has proven remarkably effective for optimization problems across machine learning and even traditional tasks in geometry processing. At the same time, the development of equivariant methods, which preserve their output under the action of rotation or some other transformation, has proven to be important for geometry problems across these domains.** In this work, we observe that naively applying Adam to optimize vector-valued data is not rotation equivariant, due to per-coordinate moment updates, and in fact this leads to significant artifacts and biases in practice. We propose to resolve this deficiency with VectorAdam, a simple modification which makes Adam rotation-equivariant by accounting for the vector structure of optimization variables. **We demonstrate this approach on common geometric optimization problems in traditional geometry processing and machine learning**, showing that equivariant VectorAdam resolves the artifacts and biases of traditional Adam when applied to vector-valued data, with equivalent or even improved rates of convergence.
> > >
> > > We have updated the abstract, and also edited some of our similarly ambiguous wording in the paper to clarify our intention (see green text in Introduction and Conclusion).
> > >
> > >
> > > ### On Preliminary Experiment
> > > _____
> > > Our current focus for VectorAdam is coordinate-valued geometric data because we found the properties of VectorAdam most evident and well-founded in theory in this realm, e.g. the proof in Sec 4.1. The adversarial optimization example, therefore, serves as a good example of optimizing geometric data in the machine learning context.
> > >
> > > However, we are also curious about the potential of applying our method to network weight optimization. In the preliminary experiment mentioned above, we used VectorAdam to optimize PointNet weights in a point cloud segmentation task. These weights are not coordinate-valued, but can be potentially interpreted geometrically as they map the point coordinates into latent spaces to perform certain geometry-aware tasks such as shape classification and segmentation. More specifically, we apply VectorAdam’s equivariant update to the rows of PointNet kernel gradients. For example, consider the kernel weights in the first convolutional layer in the shape of m x 3, which transforms the 3-dimensional point cloud coordinate data into m-dimensional latent vectors; our VectorAdam preserves the direction of the gradient vector for each row vector in the kernel. We plotted the test accuracy and loss throughout the training process, and observed a significant increase in stability using VectorAdam.
> > >
> > > We have preliminarily added this plot to the future work section for the sake of this discussion (see green text in Section 6 ). However, since this is a different setting from the coordinate optimization studied in this work which we do not deeply investigate, we want to focus on equivariant optimization of coordinate-valued data, which is more well-founded in theory as we already present in the paper. That being said, if the reviewers feel it adds value, we can include it as a preliminary experiment in the final paper which may inspire future work.

---

> > > > ### Comment · Reviewer_5HeJ · 2022-08-08
> > > > **Increase score**
> > > >
> > > > Thank you very much for your clear reply. The updated abstract does indeed more clearly separate the concepts and is, as such, more appropriate in my opinion. The preliminary experiment once again also highlights the importance of making this separation.
> > > >
> > > > Based on the previous general response of the authors, and the updated abstract, I am happy to increase my score.

---

### Author Response · Authors · 2022-08-02
**General Response 1**

We thank all reviewers for their thoughtful reviews of this submission, and we are generally glad to incorporate the suggested improvements throughout.

To briefly recap, this work presents VectorAdam, a variation of Adam optimizer that extends its scalar-based operation to vectors. Our work points out the coordinates-dependent biases introduced by Adam in geometry problems, and proposes VectorAdam as the rotation equivariant solution that outperforms or achieves similar performance as Adam in a number of experiments in geometry machine learning and optimization.

We’ll address a few common themes from reviews below, and give more detail in responses to individual reviewers. Please also see the green text in our latest paper for updates based on reviewers' suggestions.

### **1 Our paper is relevant to the NeurIPS community**
------

One concern raised in several reviews is whether this topic is relevant to NeurIPS and the context in machine learning. Beyond the fact that multiple reviewers recommend acceptance, we note that our submission is directly relevant to the thriving subcommunity of 3D geometric learning and provides insights that could inspire future work more generally across the NeurIPS community. Additionally, we note that “Optimization” is directly listed as a relevant topic in the NeurIPS call for papers.

For example, these are examples of recent papers at NeurIPS which, like this work, offer technical advances across geometry, 3D applications, and optimization in a way that is of interest to the machine learning community:

[1] Le Lidec, Quentin, et al. "Differentiable rendering with perturbed optimizers." NeurIPS 2021.

[2] Yang, Guandao, et al. "Geometry processing with neural fields." NeurIPS 2021.

[3] Yariv, Lior, et al. "Volume rendering of neural implicit surfaces." NeurIPS 2021 (Oral).

[4] Rozumnyi, Denys, et al. "Shape from blur: Recovering textured 3d shape and motion of fast moving objects." NeurIPS 2021

[5] Remelli, Edoardo, et al. "Meshsdf: Differentiable iso-surface extraction." NeurIPS 2020.

(these citations and others are enumerated at the bottom of this text)

The work in [1] is particularly similar in that it likewise presents a general-purpose optimization technique. In addition, before the prevalence of neural networks, NeurIPS has historically been a welcoming venue for many non-learning geometry-related work such as [7] and [8] that provided important insights for current learning-oriented NeurIPS work.

Within the geometric deep learning context, we also see the increasing use of classic geometry processing and optimization modules in the learning pipeline; the coordinate-dependence of these techniques makes our approach directly applicable. For instance, the Laplacian example in Figure 4 is widely used as a regularizer in geometric learning [4].

To give more examples, many existing geometric learning works directly optimize geometric data as an intermediate step within their learning pipeline. For example, [9] found that a second-stage geometry direct optimization within their neural pipeline helps to recover high-frequency details for their 3D surface reconstruction tasks; [6] directly optimizes 3D mesh vertices using a neural style transfer model for stylized surface editing. [11] also exemplified this type of geometric learning work by leveraging biharmonic weights to generate 3D deformation, for which one of our mentioned related work [13] estimates such interpolation weights in almost real-time with an Adam optimizer. We believe that this trend of combining classic geometry optimization with a learning pipeline opens up many more potential applications that can benefit from VectorAdam and the insight it brings. In addition, we also observe a line of work that samples 3D points as intermediate supervision such as [11] and [12], which might potentially also benefit from direct geometric optimization using VectorAdam. These works demonstrate direct and potential benefits of VectorAdam to ML tasks that are nontrivial in the geometric learning context. We believe that this work can inspire future advances in equivariant geometric learning and optimization.

All of this is to say that this work has great relevance and potential impact for the NeurIPS community! We are glad to incorporate this discussion into the paper text if desired.

---

> ### Author Response · Authors · 2022-08-02
> **General Response 2**
>
> ### **2 Weight-space optimization & applications beyond coordinate data**
> ------
>
> Indeed, this submission introduces the core idea of VectorAdam, and studies it mainly in the context of explicit optimization of low-dimensional coordinate quantities. In this setting the equivariance properties of VectorAdam are most straightforward, and there are already many interesting properties to study and applications to explore as shown in our existing experiments.
>
> As observed in several reviews, the basic machinery of VectorAdam extends trivially to ℝⁿ and even tensors (via, e.g., Frobenius norm-based momentum), and could be applied in more general settings such as weight matrices in in/equivariant neural networks, or neural implicit representations. However, this application is not trivial, because even when a network operates on coordinate-valued inputs, that does not necessarily mean that e.g. the columns of the weight matrices should have a corresponding equivariant structure. Other possible extensions include optimizing over more general SO(n) valued data, as suggested by Reviewer kCD5. We consider all of these to be interesting areas for future investigation, and evidence that our work will be of interest for future research in the NeurIPS community.
>
> To give an interesting example, we briefly performed a preliminary experiment in which we use VectorAdam to optimize PointNet kernels that parameterize 3D point cloud data. Here, we saw that although the final train/test accuracies were roughly similar, training with VectorAdam was dramatically more stable, perhaps due to the equivariance of optimization. While the mechanism or underlying principle of this effect needs more investigation, it nevertheless speaks to the value of geometry-aware optimization—we hope to investigate it further in future work. Also, if reviewers find this experiment interesting, we’d be happy to include it in the revision.
>
> Overall, we too are excited about the prospects of applying our simple insight in VectorAdam to more general classes of data & applications---we see it as promising material for future work across the NeurIPS community!
>
> Citations
>
> [1] (NeurIPS 2021)  Le Lidec, Quentin, et al. "Differentiable rendering with perturbed optimizers."
>
> [2] (NeurIPS 2021) Yang, Guandao, et al. "Geometry processing with neural fields."
>
> [3] (NeurIPS 2021) Yariv, Lior, et al. "Volume rendering of neural implicit surfaces."
>
> [4] (NeurIPS 2021) Rozumnyi, Denys, et al. "Shape from blur: Recovering textured 3d shape and motion of fast moving objects."
>
> [5] (NeurIPS 2020) Remelli, Edoardo, et al. "Meshsdf: Differentiable iso-surface extraction."
>
> [6] (NeurIPS 2020) Jiang, Chiyu, et al. "Shapeflow: Learnable deformation flows among 3d shapes."
>
> [7] (NeurIPS 2002) Freeman, William, and Antonio Torralba. "Shape recipes: Scene representations that refer to the image."
>
> [8] (NeurIPS 1989) Kanerva, Pentti. "Contour-map encoding of shape for early vision."
>
> [9] (CVPR 2022) Munkberg, Jacob, et al. "Extracting Triangular 3D Models, Materials, and Lighting From Images."
>
> [10] (Siggraph 2018) Liu, Hsueh-Ti Derek, Michael Tao, and Alec Jacobson. "Paparazzi: surface editing by way of multi-view image processing."
>
> [11] (CVPR 2021) Liu, Minghua, et al. "DeepMetaHandles: Learning Deformation Meta-Handles of 3D Meshes with Biharmonic Coordinates."
>
> [12] (NeurIPS 2019) Atzmon, Matan, et al. "Controlling neural level sets." Advances in Neural Information Processing Systems 32 (2019).
>
> [13] (TOG 2021) Wang, Yu, and Justin Solomon. "Fast quasi-harmonic weights for geometric data interpolation."

---

### Meta-Review · Area_Chair_8ZRd · 2022-08-23

**Recommendation:** Accept
**Confidence:** Certain

**Metareview:**

Ratings: 5/8/4/7.
Confidence: 4/3/4/4.
Discussion among reviewers: Yes.

Summary: this paper introduces a variant of Adam where instead of keeping EMAs of the squared individual gradients, the algorithm keeps an EMAs of the squared L2 norm of vectors of gradients. This EMA of the squared L2 norm is then used as normalizer, making sure that the L2 norm of weight updates are normalized. This has as advantage that the algorithm becomes equivariant to rotations in parameter space, which is crucial for certain types of problems. The reviewers noted the clear presentation and the encouraging results.

There's one reviewer with a reject rating, whose main reason for rejection is that the proposed algorithm itself is simple, in the sense that it's only a small change from baseline Adam. Other reviewers disagree that this is a reason for rejection, and I agree with the other reviewers here. Simplicity isn't bad if the method is novel, the motivation is clear, the results are good, and the empirical results are strong.

Recommendation: I recommend to accept this paper.

**Award:**

No

---

### Decision · Program_Chairs · 2022-09-14

Accept